# Loss of Glutathione-S-Transferase Theta 2 (GSTT2) Modulates the Tumor Microenvironment and Response to BCG Immunotherapy in a Murine Orthotopic Model of Bladder Cancer

**DOI:** 10.3390/ijms252413296

**Published:** 2024-12-11

**Authors:** Mugdha V. Patwardhan, Toh Qin Kane, Edmund Chiong, Juwita Norasmara Rahmat, Ratha Mahendran

**Affiliations:** 1Department of Surgery, Yong Loo Lin School of Medicine, National University of Singapore, Singapore 119228, Singapore; mugdha.p@u.nus.edu (M.V.P.); surce@nus.edu.sg (E.C.); juwita.r@nus.edu.sg (J.N.R.); 2Genomics and Data Analytics Core, Cancer Science Institute of Singapore, National University of Singapore, Singapore 117599, Singapore; kane9530@hotmail.com; 3Department of Urology, National University Hospital, National University Health System, Singapore 119074, Singapore

**Keywords:** glutathione S-transferase theta 2, urinary bladder neoplasms, bladder cancer, BCG, immunotherapy, single-cell gene expression analysis, PD-L1, inflammation

## Abstract

Loss of the glutathione-S-transferases Theta 2 (Gstt2) expression is associated with an improved response to intravesical *Mycobacterium bovis*, Bacillus Calmette-Guérin (BCG) immunotherapy for non-muscle-invasive bladder cancer (NMIBC) patients who receive fewer BCG instillations. To delineate the cause, Gstt2 knockout (KO) and wildtype (WT) C57Bl/6J mice were implanted with tumors before treatment with BCG or saline. RNA was analyzed via single-cell RNA sequencing (scRNA-seq) and real-time polymerase chain reaction (RT-PCR). BCG induced PD-L1 expression in WT mice bladders, while pro-inflammatory TNF-α was upregulated in KO bladders. ScRNA-seq analysis showed that Gstt2 WT mice bladders had a higher proportion of matrix remodeling fibroblasts, M2 macrophages, and neuronal cells. In KO mice, distinct tumor cell types, activated fibroblasts, and M1 macrophages were enriched in the bladders. In WT bladders, the genes expressed supported tumorigenesis and immunosuppressive PD-L1 expression. In contrast, Gstt2 KO bladders expressed genes involved in inflammation, immune activation, and tumor suppression. An 11-gene signature (Hmga2, Peak 1, Kras, Slc2a1, Ankfn1, Ahnak, Cmss1, Fmo5, Gphn, Plec, Gstt2), derived from the scRNA-seq analysis predicted response in NMIBC patients (The Cancer Genome Atlas (TCGA) database). In conclusion, our results indicate that patients with WT Gstt2 may benefit from anti-PD-L1 checkpoint inhibition therapy.

## 1. Introduction

Bladder cancer is the ninth most common cancer worldwide [1]. The majority of bladder cancer cases are non-muscle-invasive bladder cancers (NMIBC) [2], characterized by frequent recurrences that may progress to muscle-invasive disease. The need for continuous surveillance and therapeutic management makes the economic burden of bladder cancer high [3]. The recommended therapy for high-grade NMIBC is repeated intravesical instillations of *Mycobacterium bovis*, Bacillus Calmette-Guérin (BCG) [4]. BCG is given as an induction course for six weekly instillations, followed by a maintenance therapy of weekly BCG instillations for 3 weeks, every 3 months, for up to a year [5]. BCG immunotherapy is associated with a 30–50% failure rate, the reason for which is unknown [6]. Intravesical BCG induces a strong local immune response, mediated by reactive oxygen species production that results in the eradication of tumor cells as a bystander effect. Hence, immune and oxidative stress response gene variations may contribute to disease recurrence. Single nucleotide polymorphisms (SNPs) in genes involved in tumorigenesis (p53), inflammatory mediators (IL-6, IL2RA, IL17A, IL17RA, TNFA, IL18R1, CCR2), immune activation (NRAMP1, TRAILR1, FASL), redox-enzymes (GPX1), cytokine genes, and those associated with immune cell activation are associated with the response to BCG immunotherapy [7,8,9]. Additionally, the elevated infiltration of CD4^+^ and CD8^+^ T cells is associated with improved outcomes. In contrast, the reverse is true for suppressive cells such as M2 macrophages and regulatory T cells (Treg) [10,11]. While several predictive markers have been discovered, none are definitive, partly due to the differing patient populations, the various BCG strains used, and the therapeutic schedules followed worldwide [12].

Recently, the BCG stimulation of bladder cancer cell lines was found to increase the expression of glutathione-S-transferase theta 2 (Gstt2), a member of the glutathione-S-transferase (GST) family [13]. Patients who received less than the standard “6 + 3” BCG instillations responded better if they did not express Gstt2 [14]. The loss of Gstt2 expression results from a common 37-kilobase pair deletion in the promoter region [15]. This deletion occurs in 25% of Asians and 40% of Caucasians [15]. Previously, the Gstt2 deletion was found to be protective for the development of esophageal squamous cell carcinoma in South African (black and mixed race) populations [16]. Gstt2 plays a role in the activation of immune cells and response to infection in plants [17,18] and invertebrates [19,20]. In in vitro studies, immune cells from wildtype (WT) mice and those with the Gstt2 exon deletion (knocked out, KO) displayed differential responses to BCG attributed to the disparity in intracellular BCG survival between the WT and KO [14]. Hence, we hypothesized that Gsst2 deletion results in cellular and molecular alterations in the tumor microenvironment (TME), resulting in beneficial BCG responses.

This study aims to dissect the impact of Gstt2 expression on the response to BCG immunotherapy in tumor-bearing bladders. Gstt2 WT and KO C57BL/6J mice were orthotopically implanted with the murine bladder cancer cell line MB49-PSA. Mice were treated with fewer BCG instillations to ensure that a snapshot of the TME could be captured during the treatment course. Tumor growth characteristics and gene expression in the bladder were evaluated, and single-cell RNA sequencing (scRNA-seq) was performed to better delineate the changes in the cells in the TME. Several genes were found to correlate with differential Gstt2 expression and the observed BCG response. Finally, we interrogated the Cancer Genome Atlas (TCGA) database to determine if the genes identified in the murine tumor model correlated with outcomes, such as disease-free survival or overall survival in human bladder cancer patients with and without Gstt2 expression. 

## 2. Results

### 2.1. Tumor Growth Characteristics of the Orthotopic Model 

MB49 cells modified to secrete the human prostate-specific antigen (PSA) were orthotopically implanted in mice bladders (Figure 1A). One week after tumor implantation, mice received intravesical BCG immunotherapy (Figure 1A). Spot urine samples were collected weekly to monitor tumor growth, and PSA secretion in urine (normalized to urinary creatinine) was measured (Figure 1B,C). Mice were given four weekly BCG instillations, which is lower than the clinical schedule of six weekly instillations with booster doses. The reduced BCG instillations enabled the study of the TME during the treatment course and ensured that tumors were still present in the bladder for analysis. In untreated mice, tumors appeared to be larger in KO than in WT mice (Figure 1B,C), which was also observed upon comparison of bladder weights (Figure 1D). With BCG treatment, both WT and KO mice showed a drop in urinary PSA/creatinine secretion (Figure 1B,C), but the decline was more pronounced in KO mice (Figure 1B,C). Differences in bladder weight between untreated and BCG-treated mice correspond with differences in urinary PSA in both WT and KO mice (Figure 1D). 

RNA was extracted from whole bladders, and the expression of the human PSA gene was detected and quantified by RT-qPCR as a representation of tumor size (Figure 1E). Compared to untreated WT mice, untreated KO mice had smaller tumors (Figure 1E) despite the higher gross bladder weight (Figure 1D) and higher urinary PSA levels (Figure 1B,C). While untreated KO mice had smaller tumors (Figure 1E), the high urinary PSA levels observed in (Figure 1C) could be a result of the large tumors in the KO control group, which had a higher average RQ level, consequently leading to a large variance and standard deviation in the urinary PSA/creatinine levels at 4 weeks. BCG-treated KO mice had fewer large tumors than BCG-treated WT mice (Figure 1E). BCG therapy led to tumor absence (cured) in a small proportion of WT and KO mice (Figure 1E). 

### 2.2. Gstt2-Associated Gene Expression in the Bladder 

Successful response to BCG therapy largely depends on mounting a robust immune response [12]. Thus, the expression of key inflammatory and immune-related genes was assessed in the bladder to capture differences in immune activation and inflammation in the TME of WT and KO mice (Figure 2, Appendix A). Acute phase inflammatory cytokines IL-6, TNF-α, and IL-1β, as well as the anti-inflammatory cytokine IL-10, were measured to determine the inflammatory response to BCG. The expression of pro-inflammatory cytokines increased in both WT and KO mice after BCG treatment. However, the increase in TNF-α expression was significant in KO but not WT mice (Figure 2A). BCG significantly increased the expression of the T cell exhaustion marker, PD-L1, in WT but not in KO mice (Figure 2B). Without BCG treatment, the PD-L1 in untreated WT and KO is similar, but the TNFa appears elevated in KO versus WT mice, although not statistically significant. This observation is consistent with an inflamed environment. The significance levels were also not observed, probably due to the variance in tumor sizes within a group. Thus, the loss of Gstt2 could influence T cell activation in the TME in response to intravesical BCG via the expression of inflammatory cytokines and immune checkpoint markers.

### 2.3. Single-Cell RNA Sequencing of Bladders from WT and KO Mice 

Single-cell RNA sequencing was conducted to identify cells contributing to the T cell immunosuppressing environment. Single cells were isolated from bladders and barcoded using the split-pool technology (Parse Biosciences, Seattle, WA, USA). Quality control (QC) determined a valid barcode fraction of 60% (Appendix A), with approximately 69% of reads uniquely aligning to the mouse genome and approximately 50% of reads uniquely aligning to the mouse transcriptome (Appendix A). The mean reads per cell were 20,000, and the median genes per cell were approximately 500 (Appendix A). Post QC, 10,000 cells were retained for analysis (WT = 3800 cells and KO = 6200 cells). The data were normalized to account for differences in cell numbers between WT and KO samples. 

A total of 13 unique cell types, including urothelial cells, tumor cells, stromal cells (fibroblasts), immune cells (macrophages), neurons, myeloid cells, and endothelial cells, were identified using standard principal component analysis (PCA) and uniform manifold approximation and projection (UMAP) (Figure 3A). Bladders from WT mice had a higher proportion of matrix remodeling fibroblasts, M2 macrophages, and neurons. In contrast, bladders from KO mice had a higher proportion of tumor cells, activated fibroblasts, and M1 macrophages (Figure 3B). However, differences in the frequency of cell types were not significant. The presence of tumor cell clusters correlated with post-harvest bladder weight (Appendix A). 

Gene expression analysis within each cell cluster revealed differences in the expression of numerous genes between the WT and KO backgrounds (Figure 4, Appendix A). Specific genes were differentially expressed between WT and KO bladders across multiple cell types (Fmo5, Sh3gl2, Mecom, Peak1, Rpph1, Gphn, Plec, Ahnak, Cmss1, Hmga2) (Appendix A). The top differentially expressed genes between WT and KO mice (Appendix A) were used to identify enriched pathways by the interrogation of the Kyoto Encyclopedia of Genes and Genomes (KEGG) database and the Molecular Signatures Database (MSigDB) (Appendix A). Pathways associated with migration, cell adhesion, neuron function, stress response, and IFN-γ signaling were enriched in bladders from WT mice (Appendix A). In KO mice, pathways associated with PI3K-Akt-mTOR signaling, hypoxia, metabolism, stromal interactions, cell division and apoptosis, immune response, migration, and metabolism were enriched (Appendix A). 

### 2.4. Verification of Gene Expression in the Orthotopic Tumor Model

The expression of select genes, differentially expressed in WT and KO mice through scRNA-seq, was assessed in vitro in MB49-PSA cells through RT-qPCR (Appendix A). BCG stimulation increased the expression of Gphn, Cmss1, Sh3gl2, Peak1, Hmga2, and Gsta4, and decreased the expression of Plec, Slfn4, Rora, Saa3, Ankfn1, Slc2a1, Ahnak, Kras, Mecom, and Gstt2 in MB49-PSA cells (Appendix A). The differences were minimal except for Sh3gl2, which was markedly increased in response to BCG (Appendix A).

Differentially expressed genes were verified on another batch of mice (Table 1, Appendix A). In untreated mice, the expression of Hmga2, Peak1, Kras, Slc2a1, Gphn, Plec, and Rpph1 was higher in KO compared to WT mice, whereas the expression of Gsta4, Mecom, Sh3gl2, Ahnak, and Fmo5 was higher in WT mice (Table 1, Appendix A). In BCG-treated mice, only the expression of Slc2a1 was slightly higher in KO compared to WT mice, whereas the expression of Gsta4, Mecom, Sh3gl2, Ahnak, Cmss1, Fmo5, Gphn, Plec, and Rpph1 was higher in WT mice. The differences were not significant, likely due to the variability of the size of the tumors.

Additionally, when compared to MB49-PSA cells alone (Appendix A), the effect of BCG stimulation on some genes differed within the bladder TME (Table 1). This highlights the role of other cell types, such as stromal or immune cells, in influencing gene expression. 

### 2.5. Evaluation of Genes in Human Samples

To determine whether the genes identified by the scRNA-seq of mouse bladders could be translated to patients, the expression of genes was assessed by the interrogation of The Cancer Genome Atlas (TCGA) database using the Gene Expression Profiling Interactive Analysis (GEPIA) software version 2 (Figure 5) [21]. Of the genes selected, Gsta4 (R = 0.67, *p*-value = 0.02) and Slc2a1 (R = 0.57, *p*-value = 0.03) were most strongly correlated with Gstt2. The expression of each gene was assessed for its impact on overall survival (Figure 5A) and disease-free survival (Figure 5B). Gstt2, Gphn, Plec Cmss1, Ahnak, Slc2a1, Gsta4, Kras, Peak1, and Hmga2 were associated with worse overall survival in bladder cancer patients, and the association was significant for Gstt2 and Ahnak (Figure 5A). Ankfn1, Gstt2, Plec, Gphn, Fmo5, Cmss1, Ahnak, Mecom, Slc2a1, Gsta4, Kras, Peak1, and Hmga2 were associated with worse disease-free survival in bladder cancer patients (Figure 5B). Combinations of the abovementioned genes were selected to form gene signatures, and clinical outcomes were assessed in bladder cancer patients (Appendix A). The combination of Gstt2 and Ahnak significantly influenced overall survival in bladder cancer patients with non-papillary tumors (*p*-value = 0.025). However, the effects of any additional genes were not significant (Appendix A).

An 11-gene signature (Hmga2, Peak 1, Kras, Slc2a1, Ankfn1, Ahnak, Cmss1, Fmo5, Gphn, Plec, Gstt2) had the most substantial impact on disease-free survival in bladder cancer patients (Appendix A, Figure 5C–E). Compared to bladder cancer patients with a high expression of all 11 genes, those with low expression had a 2-fold higher disease-free survival rate (*p*-value = 0.000068) (Figure 5C). Additionally, upon segregation of bladder cancer patients by tumor type, a high expression of the 11 genes was associated with significantly lower disease-free survival in patients with both papillary (Hazard Ratio (HR) = 3.1, *p*-value = 0.0019) (Figure 5D) and non-papillary tumors (HR = 1.8, *p*-value = 0.0042) (Figure 5E). The hazard ratios were higher than that observed for each gene individually (Appendix A), suggesting a potential cumulative effect. Thus, the genes identified in the orthotopic murine model influence disease-free survival in bladder cancer patients.

## 3. Discussion

The loss of Gstt2 expression resulted in marked changes in the local bladder tumor environment in response to BCG therapy. These differences likely contribute to immune activation in the tumor microenvironment, thus influencing outcomes in bladder cancer patients. The WT Gstt2 bladder environment was characterized by elevated PD-L1 expression, which could be attributed to various cell types (tumor cells, M2 macrophages, and matrix-remodeling fibroblasts) expressing PD-L1-associated genes. In contrast, KO bladders were enriched for inflammatory cell types, such as M1 macrophages, and gene expression changes associated with immune activation, including inflammatory cytokine expression and PI3k-Akt-mTOR pathway activation. An 11-gene signature identified by scRNA-seq influenced overall survival and disease-free survival in bladder cancer patients, indicating the translatability of the orthotopic model to clinical outcomes.

The TME is complex and comprises a diverse cell populace critical to pathogenesis and treatment response. Using sc-RNAseq, the cellular ecosystem in the tumor-bearing mice bladder was dissected after BCG immunotherapy. A total of 13 cell types were identified of which two were distinct clusters of tumor cells (Figure 3). The ‘tumor cells 1’ cluster had the gene expression characteristic of stem-like tumor cells with an elevated expression of genes involved in cell division and chromosome segregation such as Top2a, Mki67, Cenpf, Cenpe, Incenp, Anln, Ccna2, Kif20b, Ccnb1, and Smc4 (Figure 3A) [22]. In contrast, the ‘tumor cells 2’ cluster was characterized by genes associated with protein synthesis (Cmss1, Rpph1, Rps12, Lars2, Ncl2), tumorigenesis (Mt-Rnr2, Hmga2, Mab1b) and cell–cell communication (Gphn, Camk1d, Il31ra, Cmss1, Rpph1) (Figure 3A), which may be indicative of an inflammatory phenotype. For example, Rpph1 is enriched in bladder cancer and linked to immune-related pathways, such as IL-1, TNF-α, Il-17 signaling, and M2 macrophage polarization [23,24]. Hmga2 also induced M2 macrophage polarization in colorectal cancer by upregulating STAT3 [25]. Both tumor cell types are more enriched in the Gstt2 KO than in the WT bladder (Figure 3B). However, the genes predominantly expressed by the tumor types in the KO bladder are involved in tumor suppressor activities such as Gphn, which reduced mTOR pathway activation, and Ahnak, which can inhibit tumor cell proliferation and invasion as demonstrated by in vitro studies (Figure 4) [26,27]. Ribosomal proteins and Jund are predominantly expressed in the WT tumor cell types, indicating a tumorigenic-leaning environment (Figure 4) [28,29,30].

One of the hallmarks of cancer is the ability to escape immune surveillance. PD-L1 is an immune checkpoint inhibitor that curbs the activation of critical immune cell components such as T cells, B cells, and dendritic cells (DCs) [31]. In Gstt2 WT mice, PD-L1 gene expression was significantly elevated after intravesical BCG treatment but not in KO mice. Sc-RNAseq analysis reveals a significant enrichment of genes associated with PD-L1 expression in the tumor cell cluster of WT mice. Jund expression increased bladder cancer cell migration and elevated PD-L1 in previous reports [32,33]. The intra-tumoral expression of PD-L1 is associated with poor outcomes in bladder cancer patients, attributed to poor T cell infiltration [34]. Hence, poorer BCG responses in WT mice and patients could result from PD-L1 upregulation in the tumor mass.

A higher M1 to M2 macrophage ratio is associated with a favorable prognosis, and the presence of M1 macrophages is associated with an improved BCG response in bladder cancer therapy [35,36]. In this study, the Gstt2 KO BCG-treated bladders displayed a higher proportion of M1 macrophages, while the WT group conversely exhibited a more enriched M2 macrophage population. Enriched genes in the M1 macrophages of Gstt2 KO bladders, such as Peak1, Gphn, and Alcam (Figure 4), are associated with signaling pathways, such as the ERK/MAPK pathway, cell proliferation, and T cell activation [37,38,39,40]. An increased expression of such proteins in M1 macrophages may be involved in lineage expansion and T cell interactions, leading to an activated inflammatory immune response in the TME. The M1 macrophages may also contribute to the significantly higher TNF-α expression in KO bladders (Figure 2), which is associated with a favorable BCG response [12,41,42,43]. Conversely, elevated M2 macrophages in WT bladders can contribute to immune exhaustion by suppressing T cell activity in response to BCG treatment [11,35,44]. The M2 macrophage cluster in WT mice was enriched to express inflammation-associated genes such as Saa3, Ccl5, Ly6a, and C3 (Figure 4). Saa3 expression in murine macrophages has been linked to the increased suppressive activity of myeloid-derived suppressor cells (MDSCs) [45]. Although Ccl5 is a T cell chemoattractant, its expression in macrophages was associated with PD-L1 levels, leading to immunosuppression [46].

Fibroblasts are the arbiters of extracellular matrix (ECM) remodeling and influence the pathogenesis of organs in disease states, such as epithelial to mesenchymal transition (EMT) during tissue fibrosis and cancer progression [47,48]. Bladder neuronal cells are responsible for the muscular control of bladder functions, such as urine storage and voiding [49]. The proportions of matrix remodeling fibroblasts and neuronal cells are more enriched in the WT bladders after intravesical BCG (Figure 3B). The genes expressed by the matrix remodeling fibroblasts (Cd55, Cxcl10, and Ccl2) in WT bladders are associated with increased collagen synthesis [50,51], survival [52], invasion [53], and immunosuppression [54] (Figure 4). Collagen deposition in the bladder correlated with cancer cell growth and progression [55], resulting in ECM stiffness, which can impair CD8+ T cell penetration [56] and increase PD-L1 expression [57]. Hence, the matrix remodeling fibroblasts could mold an immunosuppressive TME conducive to tumor growth. The reason for the increased frequency of neuronal cells in the WT bladders is unclear (Figure 3B). However, the neuronal cells in Gstt2 KO bladders express genes that protect them from ferroptotic (Fth1 and Ftl1) and pyroptotic (Ybx1) cell death [58,59,60] (Figure 4). It is plausible that the protection from neuronal cell death in the Gstt2 KO maintains bladder function.

In this study, scRNA-seq helped delineate the cellular environment of the bladder after BCG therapy. However, the technique has limitations as it uses a reference marker gene set, so rare or new cell types may be missed. Furthermore, the mean reads per cell and median genes per cell were lower than optimal, which may have impacted the resolution of cell clusters and the detection of rare cell types. The interpretation of cell clusters with relevant biological characteristics is challenging and must be handled with caution due to biological and technical variations associated with the procedure [61]. Additionally, the validation of genes in the orthotopic tumor model was carried out by RT-qPCR assessment of whole bladders. The use of whole bladders may have diluted the transcripts of interest in the tumor cells confined to the inner urothelial lining. Hence, similar work in the future should consider isolating cells on the bladder urothelium by exposing the bladders and scraping the inner bladder wall for sample retrieval. Nevertheless, scRNA-seq has provided valuable insights into the events in the bladder TME post-BCG therapy. The clinical association of an 11-gene signature with overall and disease-free survival indicates the translatability of the orthotopic model. Such gene signature data may allow physicians to identify patients who may not respond well to BCG immunotherapy. Furthermore, the results suggest that patients with WT Gstt2 expression may benefit from a combination of BCG and anti-PD-L1 checkpoint inhibition therapy.

## 4. Materials and Methods

### 4.1. In Vivo Study

The Institutional Animal Care and Use Committee of the National University of Singapore approved the breeding (BR17-1422) and animal experiment protocols (R17-1434). GSTT2KO mice were generated as previously described [14]. GSTT2 knockdown was performed by the Transgenic Mice Facility, Cancer Science Institute (CSI), using the CRISPR system to delete exon 2 of the murine GSTT2 gene. Heterozygous mice were mated to generate WT and KO mice. WT and KO female C57BL6/J mice (4–6 months old) were anesthetized with an intraperitoneal anesthetic formulation (ketamine (75 mg/kg) and medetomidine (1 mg/kg)) and given 100 µL of compound sodium lactate solution intraperitoneally (B. Braun, Penang, Malaysia). Duratears eye ointment was applied. Spot urine samples were collected by gently palpating the lower abdomen. The bladders were primed by orthotopic instillation of Poly-L-lysine (PLL) solution (Sigma-Aldrich, St. Louis, MO, USA) (50 µL of 0.01% PLL) using a 24G catheter (B. Braun, Penang, Malaysia). The PLL solution was maintained in the bladder for 30 minutes, after which the catheter was removed, and the bladder was pressed gently to remove any excess solution. MB49-PSA cells (1 × 10^5^ cells in 50 µL of blank DMEM) were instilled orthotopically through a 24G catheter and maintained in the bladder for 1 h, after which the bladder was pressed gently to remove any excess solution [62]. The anesthetic was reversed by intraperitoneal atipamezole (1 mg/kg).

The BCG treatment was carried out similarly. Mice were anesthetized, and urine samples were collected, as described above. BCG (3 × 10^6^ CFU in 100 µL) was instilled orthotopically through a 24G catheter and maintained in the bladder for 1 h, after which the bladder was pressed gently to remove any excess solution. The anesthetic was reversed as per above. Mice were given weekly saline or BCG installations for 4 weeks and were culled one day after the fourth instillation. Whole bladders were harvested in either RNA-later^TM^-ICE (Thermo Fisher, Waltham, MA, USA) for RT-qPCR analysis (Appendix A) or in DMEM blank media for sc-RNAseq.

### 4.2. Analysis of the Human TCGA Database

The gene expression profiling interactive analysis 2 (GEPIA2) software version 2 was used to analyze the Cancer Genome Atlas (TCGA) database. The ‘survival map’ and ‘survival analysis’ functions were used. Gene expression was normalized to Rps27a. The analysis was performed on 15 October 2024.

### 4.3. Statistical Analysis

All analysis was conducted on Prism GraphPad version 9 (RRID: SCR_002798). One-way ANOVA was carried out to compare between groups with Tukey’s post hoc test. A *p*-value < 0.05 was considered to indicate a statistically significant difference.

## 5. Conclusions

The effects of Gstt2 deletion were broad and influenced various cell types and genes within the TME that can drive favorable BCG responses via activating inflammatory immune responses. An 11-gene signature identified by scRNA-seq significantly impacted clinical outcomes in bladder cancer patients, highlighting the translatability and relevancy of the orthotopic model. Gstt2 deletion may exert a pro-inflammatory and antitumorigenic effect in response to BCG via the PI3K-Akt-mTOR pathway. However, this needs to be validated in future studies. Patients with WT Gstt2 expression may benefit from anti-PD-L1 immune checkpoint inhibition therapy but not Gstt2 KO patients.

## Figures and Tables

**Figure 1 ijms-25-13296-f001:**
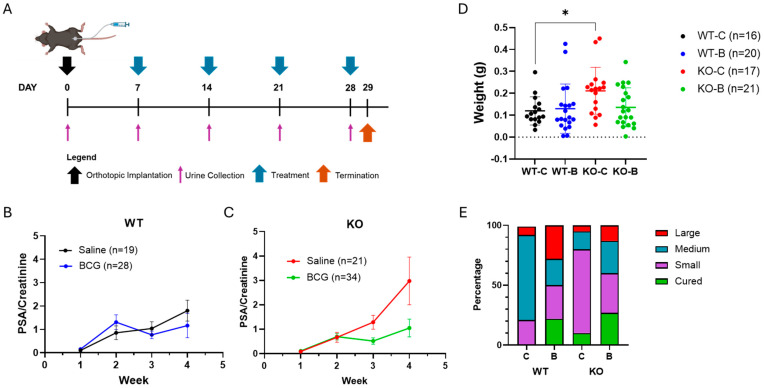
Tumor growth in the orthotopic model. (**A**) The murine bladder cancer cells expressing human prostate-specific antigen (MB49–PSA) were implanted orthotopically (1 × 10^5^ cells) into Gstt2 WT and KO C57BL6/J female mice. One week post-implantation, the mice were treated weekly with either saline or 3 × 10^6^ colony forming units (CFU) BCG instillations for 4 weeks. Urine samples were collected weekly, and secreted urinary PSA normalized with creatinine levels was assessed by ELISA for (**B**) WT and (**C**) KO mice. Error bars represent the standard error of the mean (SEM). (**D**) Bladders were harvested one day after the fourth BCG instillation and weighed; error bars represent standard deviation, and statistical significance was determined using one–way ANOVA (* *p*–value < 0.05). (**E**) Quantitative real–time PCR was used to measure PSA expression in whole bladders (WT–C: n = 14, WT–B: n = 18, KO–C: n = 20, KO–B: n = 16). The relative quantification (RQ) values were used to categorize tumors as small (0.0 < RQ ≤ 0.5), medium (0.5 < RQ ≤ 1.5), or large (RQ ≥ 1.5). Mice were cured if the cycle threshold (CT) value >35. WT—Wildtype, KO—Knockout, C—Control Saline, B—BCG.

**Figure 2 ijms-25-13296-f002:**
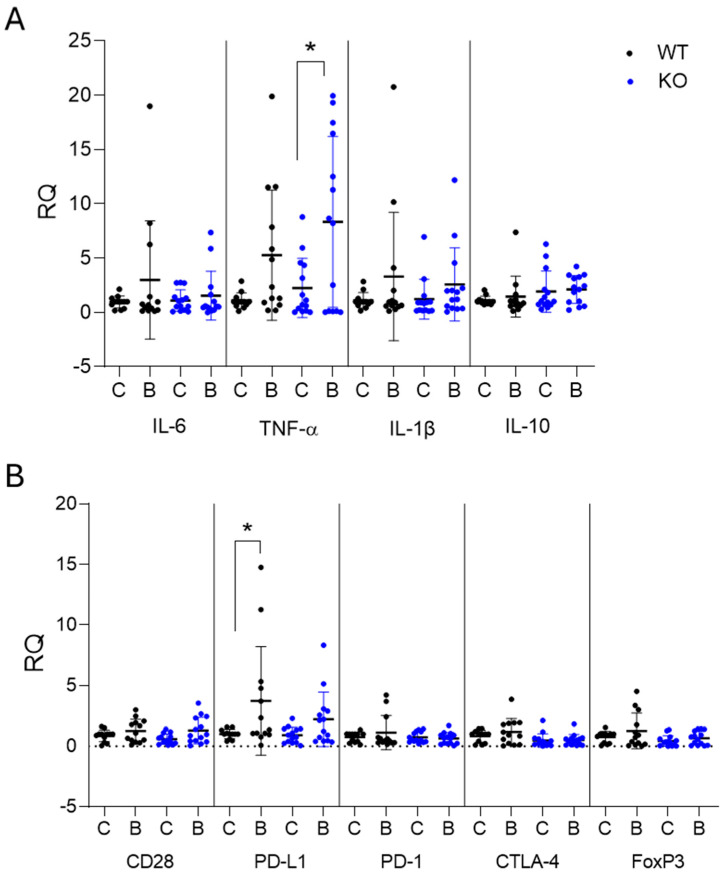
Expression of inflammatory and exhaustion-associated genes in bladders from WT and KO mice. MB49–PSA bladder cancer cells were implanted orthotopically into Gstt2 WT and KO C57BL6/J female mice. The mice were treated with weekly BCG instillations one week post-implantation for 4 weeks. Bladders were harvested one day after the fourth BCG instillation, and RNA was extracted for quantitative real–time PCR. (**A**) The expression of cytokines and (**B**) immune activation and exhaustion markers was compared in WT and KO mice. Comparisons of means were performed using one–way ANOVA, and the mean of each group was compared to the mean of every other group using Tukey’s test for multiple comparisons. * Significant differences were observed when comparing control to BCG–treated mice (* *p*–value < 0.05). WT—Wildtype, KO—Knockout.

**Figure 3 ijms-25-13296-f003:**
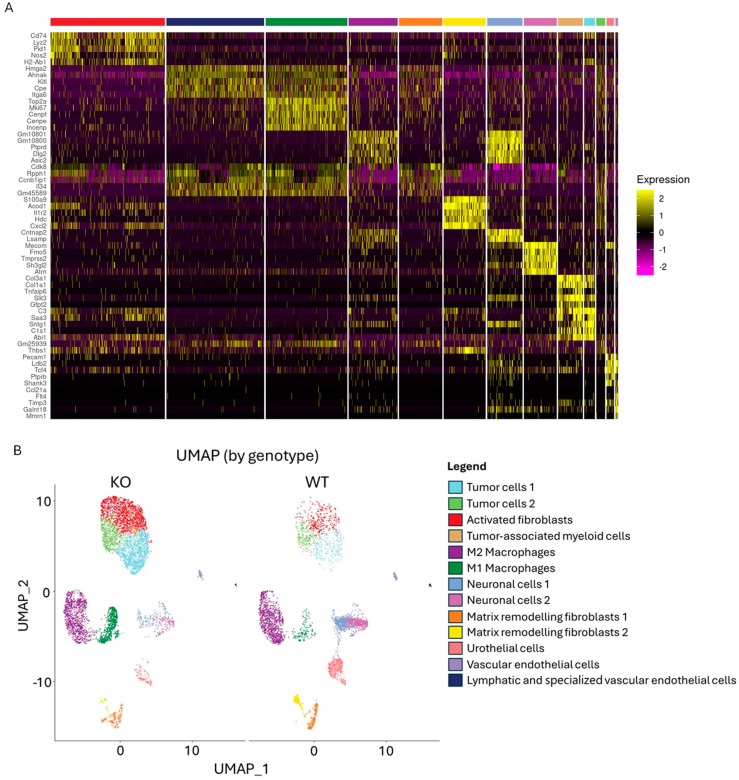
Single–cell RNA sequencing of whole bladders from WT and KO mice. MB49–PSA bladder cancer cells were implanted orthotopically into Gstt2 WT and KO C57BL6/J female mice. The mice were treated with weekly BCG instillations one week post-implantation for 4 weeks. Bladders were harvested one day after the fourth BCG instillation, and single cells were isolated for single–cell RNA sequencing. (**A**) Standard principal component analysis (PCA) and uniform manifold approximation and projection (UMAP) were used to cluster cells, and the top differentially expressed genes between the clusters were used to determine cell type. (**B**) The UMAP plots were segregated by genotype to compare differences in the frequency of cell types between WT and KO mice. WT—Wildtype, KO—Knockout.

**Figure 4 ijms-25-13296-f004:**
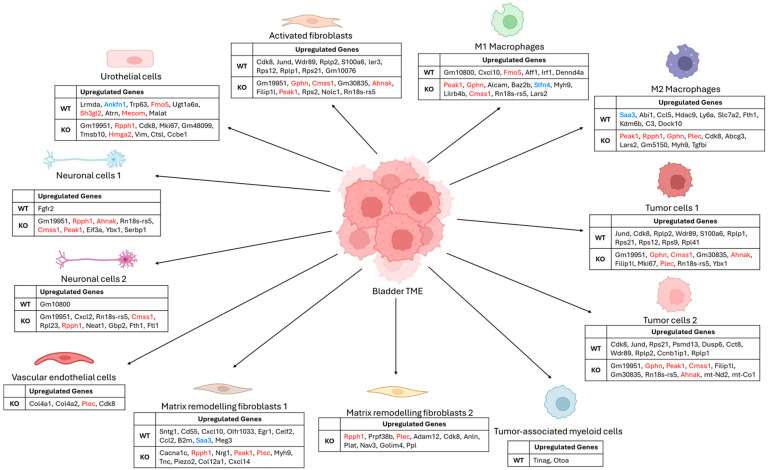
Differentially expressed genes between WT and KO backgrounds within individual cell clusters of the bladder TME. The “FindAllMarkers” function was used to identify differentially expressed genes (DEG) within each cell cluster between WT and KO backgrounds. In total, 12 clusters with differentially expressed genes were identified based on log2(Fold Change) > 0.5 and *p*-value < 0.05. The top 10 differential genes for each cell cluster are presented (from highest to lowest log2(Fold Change)). The genes in the blue font were validated in MB49–PSA cells, and those in the red font were selected for validation in both MB49–PSA cells and the murine model. WT—Wildtype, KO—Knockout.

**Figure 5 ijms-25-13296-f005:**
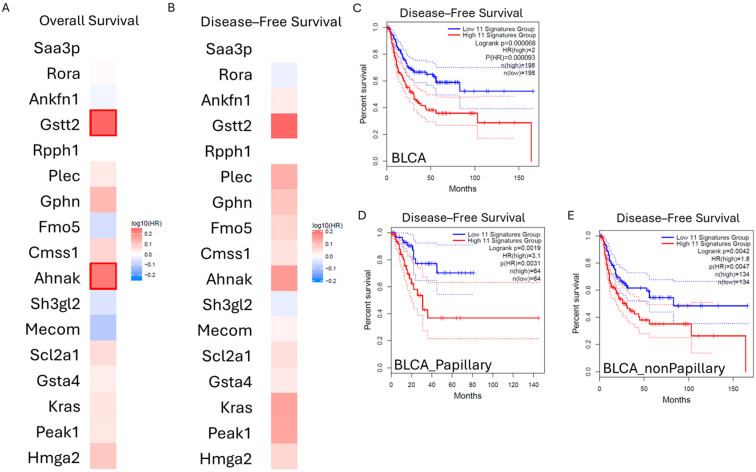
Correlation of select genes with clinical outcomes in bladder cancer patients. The expression of the genes identified by sc–RNAseq was compared in bladder cancer patients by interrogation of the TCGA database using the GEPIA software version 2 [21]. The hazard risk associated with each gene with respect to (**A**) overall survival or (**B**) disease–free survival is shown: red = increased risk, blue = decreased risk; outlined boxes = significant. (**C**) Eleven genes (Ankfn1, Gstt2, Plec, Gphn, Fmo5, Cmss1, Ahnak, Slc2a1, Kras, Peak1, Hmga2) were selected to form a gene signature. Disease–free survival was assessed with respect to the 11 genes (normalized with RPS27a) in bladder cancer patients and by bladder cancer sub–type, namely (**D**) papillary and (**E**) non-papillary tumors.

**Table 1 ijms-25-13296-t001:** Validation of single-cell RNA sequencing data.

Gene	Gstt2 WT		Gstt2 KO	
	Control (*n* = 13)	BCG-Treated (*n* = 11)	Control (*n* = 13)	BCG-Treated (*n* = 11)
Hmga2	1.116 ± 1.092	1.962 ± 2.385	1.753 ± 1.803	1.817 ± 1.578
Peak1	1.118 ± 0.939	1.655 ± 1.789	1.316 ± 1.009	1.505 ± 1.477
Kras	1.337 ± 1.141	1.395 ± 1.345	1.662 ± 1.289	1.417 ± 0.762
Gsta4	1.015 ± 1.278	1.965 ± 2.838	0.431 ± 0.531	0.630 ± 0.537
Slc2a1	0.798 ± 0.374	1.346 ± 1.453	1.126 ± 1.397	1.623 ± 1.612
Mecom	1.018 ± 0.834	1.206 ± 1.041	0.535 ± 0.669	0.515 ± 0.218
Sh3gl2	1.146 ± 1.689	1.636 ± 1.905	0.565 ± 1.105	0.410 ± 0.275
Ahnak	1.535 ± 1.138	2.099 ± 2.171	1.128 ± 0.577	1.091 ± 0.445
Cmss1	0.891 ± 0.241	1.027 ± 0.835	0.875 ± 0.477	0.834 ± 0.559
Fmo5	1.514 ± 2.473	2.422 ± 3.572	0.654 ± 0.513	0.957 ± 0.757
Gphn	0.903 ± 0.369	2.525 ± 4.091	1.372 ± 1.077	2.049 ± 1.597
Plec	0.885 ± 0.432	5.124 ± 13.080	1.972 ± 1.894	3.254 ± 4.382
Rpph1	8.373 ± 23.800	8.912 ± 18.820	29.290 ± 43.470	7.992 ± 7.489

Data presented as mean relative quantification (RQ) ± SD. Comparisons of means were performed using one–way ANOVA with Tukey’s test for multiple comparisons. No significant differences were found. WT—Wildtype, KO—Knockout.

## Data Availability

All research data supporting this publication are included in the manuscript and Appendix A. The scRNAseq data that support the findings are available in the NCBI GEO database (https://www.ncbi.nlm.nih.gov/geo/) with accession number GSE281932.

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
