# Peer review of "Loss of Glutathione-S-Transferase Theta 2 (GSTT2) Modulates the Tumor Microenvironment and Response to BCG Immunotherapy in a Murine Orthotopic Model of Bladder Cancer"

_ijms, 2024, doi:10.3390/ijms252413296_

Round 1

Reviewer 1 Report

Comments and Suggestions for Authors

This manuscript aimed to study the role of Gstt2 in the tumor microenvironment in response to BCG therapy. Gene expression and cell types were compared between Gstt2 WT and KO mice with and without BCG treatment. However, many observed differences were statistically insignificant. For example, differences in the frequencies of 13 cell types were not significant, and the expression levels of all genes verified in orthotopic tumor models also showed no significant differences. These findings suggest that the study does not provide substantial insights into the role of Gstt2.

In addition, the reads per cell (20,000) and the number of genes per cell (500) are lower than optimal for many scRNA-seq applications. This might have limited the ability to robustly analyze cell states, detect rare populations, or perform differential expression analyses.

Here are some specific concerns.

In Figure 1B, comparing untreated WT and KO, PSA/Creatinine levels appear to be 1.5-fold higher in KO mice compared to WT mice. However, in Figure 1E, there are more mice with smaller tumors in the KO group compared to the WT group. Based solely on the tumor sizes in the two groups: small, medium, and large, it appears that PSA levels in KO mice would be lower than in WT mice. If so, this would conflict with the data in Figure 1B. Please explain this apparent discrepancy. 

Table 1: Are there significant differences in TNF-α and PD-L1 levels between Gstt2 WT and KO upon BCG treatment? If so, what does this imply? Another comparison, between Gstt2 WT and KO without BCG treatment, should also be performed. These comparisons could provide comprehensive insights into the role of Gstt2 in regulating the immune response.

Table 1 and Table 2: The data in these tables should be represented as bar graphs or other types of graphs to more effectively convey the results.

Line 88: “Bladder weights reflect similar differences between untreated and BCG-treated mice of WT and KO backgrounds.” This statement is unclear. What does “similar differences” refer to? Does it mean the differences in bladder weights between untreated and BCG-treated mice when comparing WT and KO backgrounds? Or does it refer to a comparison between PSA differences and bladder weight differences?

Line 93: “This may indicate greater infiltration of immune cells or differences in tumor stroma.” This sentence lacks clarity, as it does not specify which mice have greater infiltration.

Comments on the Quality of English Language

Some writing lacks clarity and essential specific information, making it difficult to understand what is being described.

Author Response

Comment 1: This manuscript aimed to study the role of Gstt2 in the tumor microenvironment in response to BCG therapy. Gene expression and cell types were compared between Gstt2 WT and KO mice with and without BCG treatment. However, many observed differences were statistically insignificant. For example, differences in the frequencies of 13 cell types were not significant, and the expression levels of all genes verified in orthotopic tumor models also showed no significant differences. These findings suggest that the study does not provide substantial insights into the role of Gstt2.

In addition, the reads per cell (20,000) and the number of genes per cell (500) are lower than optimal for many scRNA-seq applications. This might have limited the ability to robustly analyze cell states, detect rare populations, or perform differential expression analyses.

Response: The abovementioned limitation of the lower reads per cell and genes per cell has been acknowledged in the discussion section (lines 329-330): “Lower mean reads per cell and median genes per cell were lower than optimal, which may have impacted resolution of cell clusters and detection of rare cell types.” Additionally, the genes identified by scRNA-seq was verified by RT-qPCR assessment of whole bladders, including the bladder wall and the urothelial lining. Assessing whole bladders may dilute the transcripts of interest in the implanted tumors, which are confined to the urothelial lining in this model. The choice of sample collection could explain the lack of significant gene expression differences between the WT and KO. We have amended the discussion section to acknowledge the abovementioned limitations (lines 333-336). We have also included the recommendation of extracting the inner bladder lining for future analysis in the discussion (lines 350-352). A further source of variation is that tumors in mice are not all the same size so immune cell infiltration into the tumors would not be the same, leading to differences in gene expression. Further, with the number of mice evaluated, this variation would impact the likelihood of finding statistically different gene expression outcomes. Thus, we also evaluated the genes that were identified on the TCGA database, as this would confirm the validity of the findings.

Comment 2: Here are some specific concerns.

In Figure 1B, comparing untreated WT and KO, PSA/Creatinine levels appear to be 1.5-fold higher in KO mice compared to WT mice. However, in Figure 1E, there are more mice with smaller tumors in the KO group compared to the WT group. Based solely on the tumor sizes in the two groups: small, medium, and large, it appears that PSA levels in KO mice would be lower than in WT mice. If so, this would conflict with the data in Figure 1B. Please explain this apparent discrepancy.

Response: The subsets of mice used for urinary PSA analysis (Figure 1B-C) and RT q-PCR assessment of PSA expression (Figure 1E) were not similar, which may have led to discrepancies. To clarify, urinary PSA was measured for all mice. Then, a subset of the mice was used for RT-qPCR (Figure 1E), while another subset was used for scRNA-seq. To clarify this matter, the n numbers for Figure 1E have been added to the figure legend (line 112).

Another reason for the discrepancy lies in the data expression for Figure 1E. The figure presents the proportion of mice with ‘small, medium’ or ‘large’ tumors rather than the RQ values. While WT-C mice have more large tumors than KO-C mice, the average RQ values of the large tumors are higher in the KO-C group (3.8) compared to WT-C mice (3.6). These large tumors likely resulted in the high variances of PSA/creatinine levels observed in Figure 1C, where the standard deviation for KO-C is very large leading to the discrepancies between Figure 1E and Figures 1B-C. This explanation has been acknowledged in the results section (lines 94-100):

“ While untreated KO mice had smaller tumors (Figure 1E), the high urinary PSA levels observed in (Figure 1C) could be a result of the large tumors in the KO control group, which has a higher average RQ level, consequently leading to large variances and standard deviation in the urinary PSA/creatinine levels at 4 weeks ”

The sentence following this (line 96-97): “This may indicate greater infiltration of immune cells or differences in tumor stroma” has been removed.

Comment 3: Table 1: Are there significant differences in TNF-α and PD-L1 levels between Gstt2 WT and KO upon BCG treatment? If so, what does this imply? Another comparison, between Gstt2 WT and KO without BCG treatment, should also be performed. These comparisons could provide comprehensive insights into the role of Gstt2 in regulating the immune response

Response: We performed one-way ANOVA to determine statistically significant differences between means. Tukey’s multiple comparisons test was carried out to compare the mean of each group with the mean of every other group. Only the comparisons that yielded significant differences were marked with an asterisk. Information pertaining to this has been added to the figure legend (lines 137-140). A comparison of BCG-treated WT and KO mice did not reveal any significant differences in TNF-α and PD-L1 expression. Comparison between WT and KO untreated mice also showed no significant differences.

Without BCG treatment, the PD-L1 in untreated WT and KO is similar, but the TNFa appears elevated in KO versus WT mice, although not statistically significant. This observation is consistent with an inflamed environment. Additionally, the significance levels were not observed, probably due to the variance in tumor sizes within a group. The explanation for the lack of differences observed between WT and KO control samples were added in the main manuscript (Line 131-135).

Comment 4: Table 1 and Table 2: The data in these tables should be represented as bar graphs or other types of graphs to more effectively convey the results.

Response: Table 1 has been moved to the supporting information section (Supplementary Table 2), while a figure corresponding to Table 1 has been added to the main body (Figure 2). Table 2 is retained (now referred to as Table 1) since the table format, rather than the figure, is more conducive for extracting information. However, a corresponding figure has been added to the supplementary section (Supplementary Figure 5) for visual representation.

Comment 5: Line 88: “Bladder weights reflect similar differences between untreated and BCG-treated mice of WT and KO backgrounds.” This statement is unclear. What does “similar differences” refer to? Does it mean the differences in bladder weights between untreated and BCG-treated mice when comparing WT and KO backgrounds? Or does it refer to a comparison between PSA differences and bladder weight differences?

Response: This sentence has been changed to: “Differences in bladder weight between untreated and BCG-treated mice correspond with differences in urinary PSA in both WT and KO mice” (line 88-89).

Comment 6: Line 93: “This may indicate greater infiltration of immune cells or differences in tumor stroma.” This sentence lacks clarity, as it does not specify which mice have greater infiltration.

Response: This line has been removed to avoid confusion.

Reviewer 2 Report

Comments and Suggestions for Authors

Unresponsiveness to BCG instillation presents a significant clinical challenge in the treatment of bladder urothelial carcinoma (UC), particularly carcinoma in situ (CIS). Loss of glutathione-S-transferase theta 2 (GSTT2) expression has been reported to be associated with an improved response to BCG following a short course of treatment. The authors used an orthotopic UC mouse model with GSTT2 knockout (KO) and wild-type (WT) mice, which were instilled with either normal saline (NS) or BCG intravesically, to investigate the mechanism underlying this phenomenon. In addition, they performed bioinformatic analysis using the human TCGA database.

The authors found that GSTT2 KO modulated the tumor microenvironment (TME) in a way that enhanced BCG efficacy. They conclude that patients with WT GSTT2 expression, but not those with GSTT2 deletion, may benefit from immune-oncology (IO) therapies.

The manuscript is well-written and includes two tables, four figures, and an additional supplementary figure, citing 67 references. While the study is both interesting and potentially impactful for the management of UC patients, the authors should further elaborate on the specific mechanism by which GSTT2 modulates the tumor microenvironment. Also, the reason for observing this phenomenon in a short course of BCG should be mentioned

Minor point:
The phrase "less therapy" in the third line of the abstract, which appears to refer to a shorter-than-standard course of BCG therapy, is unclear and should be reworded for clarity.

Comments on the Quality of English Language

The language of the manuscript is fine.

Author Response

Comment 1: Unresponsiveness to BCG instillation presents a significant clinical challenge in the treatment of bladder urothelial carcinoma (UC), particularly carcinoma in situ (CIS). Loss of glutathione-S-transferase theta 2 (GSTT2) expression has been reported to be associated with an improved response to BCG following a short course of treatment. The authors used an orthotopic UC mouse model with GSTT2 knockout (KO) and wild-type (WT) mice, which were instilled with either normal saline (NS) or BCG intravesically, to investigate the mechanism underlying this phenomenon. In addition, they performed bioinformatic analysis using the human TCGA database.

The authors found that GSTT2 KO modulated the tumor microenvironment (TME) in a way that enhanced BCG efficacy. They conclude that patients with WT GSTT2 expression, but not those with GSTT2 deletion, may benefit from immune-oncology (IO) therapies.

The manuscript is well-written and includes two tables, four figures, and an additional supplementary figure, citing 67 references. While the study is both interesting and potentially impactful for the management of UC patients, the authors should further elaborate on the specific mechanism by which GSTT2 modulates the tumor microenvironment. Also, the reason for observing this phenomenon in a short course of BCG should be mentioned

Minor point:

The phrase "less therapy" in the third line of the abstract, which appears to refer to a shorter-than-standard course of BCG therapy, is unclear and should be reworded for clarity.

Response: This sentence has been reworded to “Loss of the glutathione-S-transferases Theta 2 (Gstt2) expression is associated with improved response to intravesical Mycobacterium bovis, Bacillus Calmette Guérin (BCG) immunotherapy for non-muscle invasive bladder cancer (NMIBC) patients who receive fewer BCG instillations.” (Line 14-15)